# Evaluation of Different Tomato (*Solanum lycopersicum* L.) Entries and Varieties for Performance and Adaptation in Mali, West Africa

Wubetu Bihon [1,2,*], Kukom Edoh Ognakossan [2], Jean-Baptiste Tignegre [2], Peter Hanson [3], Kabirou Ndiaye [2] and Ramasamy Srinivasan [3]

[1] World Vegetable Center, Eastern and Southern Africa, Addis Ababa P.O. Box 5689, Ethiopia
[2] World Vegetable Center, Western & Central Africa—Dry Regions, Samanko Research Station, Bamako BP 320, Mali; kukom.edoh@worldveg.org (K.E.O.); jean-baptiste.tignegre@worldveg.org (J.-B.T.); riz.ndiaye@gmail.com (K.N.)
[3] World Vegetable Center, 60 Yi Ming Liao, Tainan 74151, Taiwan; peterhansonmn@gmail.com (P.H.); srini.ramasamy@worldveg.org (R.S.)
* Correspondence: wubetu.legesse@worldveg.org

**Abstract:** Tomato is an important vegetable crop and plays a major role in the food and nutrition security of the people of Mali. Production has increased in the recent decades but improvement in the fruit yield and quality remains suboptimal. Limited access to the best-adapted tomato varieties to the local conditions, pests and diseases are the major limiting factors for improving productivity. This study evaluated the performance of different tomato entries and varieties for their productivity, resistance to pests and diseases and postharvest fruit quality in Mali. Twenty-two entries and varieties of tomato in the rainy season and twenty-four in the dry season were evaluated. Varieties that were well adapted, better yielded, disease resistant and with good fruit quality were identified. Major plant diseases observed included tomato yellow leaf curve disease (TYLCD), bacterial wilt, bacterial leaf spot, early blight and southern blight. However, TYLCD was the major problem during the dry season. The variety of Icrixina was the most affected by TYLCD in both the rainy and dry seasons, although its total yield was not affected and remained one of the highest. Konica was one of the most susceptible varieties to bacterial wilt and bacterial leaf spot diseases. Tomato accession AVTO1710 provided the highest fruit yield (40.9 t/ha), while AVTO1704 provided the lowest (6.50 t/ha) in the rainy season. In contrast the highest yield during the dry growing season was 20 t/ha from VIO43614. Tomato entries and varieties varied in their postharvest fruit quality attributes (firmness, total soluble solid, pH and dry matter). Production season clearly influenced yield, disease occurrence and severity, as well as postharvest fruit qualities. The study identified better disease-resistant and yielding tomato entries suitable for rainy and dry growing seasons, which can be considered and scaled up for production so that farmers in Mali can produce tomato all year round.

**Keywords:** *Solanum lycopersicum*; adaptation; growing season; fruit quality

## 1. Introduction

Tomato (*Solanum lycopersicum*) is one of the most popular, widely cultivated and consumed vegetable crops worldwide [1]. As a high value crop, tomato is a major income source for smallholder farmers [2]; it is also a rich source of micronutrients and minerals, including lycopene, beta-carotene, potassium, calcium, flavonoids, folate and ascorbic acid, which play an important role in human health [2,3]. Tomato is a cash crop and increasingly important for smallholder farmers in Mali. The crop is a widely used and versatile vegetable in the country. Tomato forms an important component of food consumed and this is evident in the fact that many dishes in West Africa have tomatoes as a component or ingredient. It can be consumed in a variety of ways, including in a fresh salad, cooked in other dishes

or processed into other food products, including tomato paste [3–5]. Per capita tomato consumption increases in the world, which can be attributed to increased use of processed tomato products, intensification of trade, increased urbanization and population. Similarly, tomato consumption in Mali has also increased year after year due to the reasons mentioned above and the rapidly emerging demand, urbanization, rising incomes and standards of living [6].

Tomatoes are one of the largest vegetable crops in Mali with a total production area that has more than doubled from 5000 ha in 2012 to 12,354 ha in 2018, and production has increased from 96,000 t/year to 204,698 t during the same period. However, productivity remains at 16–17 t/ha [7], which is less than in the neighboring countries, such as Senegal (21.3 t/ha) and Niger (25.0 t/ha) [7]. Tomatoes are best adapted to temperatures between 18 and 27 °C, and high temperatures in Mali that can reach up to 45 °C during the hot dry season result in high flower drop that leads to reduced yields. Other factors contributing to reduced tomato yields include pests and diseases, poor production skills and inaccessibility of inputs [8–10]. High postharvest losses further reduce tomato supplies to consumers.

Tomato production in West Africa in general and Mali in particular, is hampered by multiple fungal, viral and bacterial diseases. Begomoviruses vectored by the whitefly, *Bemisia tabaci* Genn. (Aleyrodidae: Hemiptera) cause tomato yellow leaf curl diseases (TYLCD) that chronically threaten tomato production in Mali and other West African countries [11]. Fungal diseases that reduce the quality and quantity of the tomato in Mali include early blight caused by *Alternaria solani*, septoria leaf spot caused by *Septoria lycopersici*, fusarium wilt caused by *Fusarium oxysporum* f sp. *lycopersici*, southern blight caused by *Sclerotium rolfsii* and verticilium wilt caused by *Verticilium dahliae*. Bacterial wilt disease caused by the *Ralstonia* species complex (RSC) is one of the most destructive diseases of tomato worldwide and in Mali as well [10,12,13]. Insect pests such as South American leaf miner, *Tuta absoluta* Meyrick (Gelechiidae: Lepidoptera) [14], fruit borer, *Helicoverpa armigera* Hübner (Noctuidae: Lepidoptera), spider mites, *Tetranychus* spp. (Tetranychidae: Trombidiformes) and disease transmitting vectors, mainly whiteflies, are major constraints in tomato production [8].

In an effort to control tomato pests and diseases, farmers in urban and peri-urban areas excessively use several groups of unlabeled synthetic pesticides such as Thiram, Mancozeb, difenoconazole, cypermethrin, etc. [15]. Tomato production in rural areas targeting production for local markets and home consumption is often left unprotected from disease and insect infestations. Use of improved varieties adapted to abiotic stresses and resistant to pests and diseases is among the least expensive and durable strategies for yield and quality improvements. Farmer use of improved, disease resistant varieties could result in reduced pesticide use which is expensive and causes environmental concerns. Tomato varieties/entries that combine the adaptation to local environment, yield potential, fruit quality and resistance to pests and diseases are strongly needed to enhance productivity, supply of nutritious food for consumers and to increase the incomes of smallholder farmers. The objective of this study was, therefore, to test a range of different tomato entries and varieties and identify comparatively better performance and adaptability to the local conditions in Mali, with better fruit quality.

## 2. Materials and Methods

### 2.1. Study Site Condition

The study was conducted at Samanko research station (12.526 N 8.068 W, 350 masl) in Bamako, Mali. Mali is located in sub-Sahelian vegetation belt in West Africa between 12° W and 4° E longitude and 10° and 25° N latitude. The annual rainfall regime is monomodal with distinct wet and dry seasons and air temperature varies greatly during the early months of the year. In the site where this study was conducted, there was no rainfall recorded in the dry cropping season between November and March and the relative humidity was between 47 to 57%. In contrary, there was rainfall from June to October with maximum rainfall of 73 mm in August. In the dry season the average monthly

maximum temperature ranged from 39 °C in November to 42 °C in March while the minimum temperature was 19 °C in January. The maximum average temperature from July to October was 37 °C. The minimum temperature was between 24 °C from June to October in the rainy season tomato production.

### 2.2. Experimental Layout

Tomato entries introduced from the World Vegetable Center (WorldVeg) tomato breeding program in Taiwan, farmer-preferred varieties grown in Mali, and Hawaii 7996 (H7996) a bacterial wilt resistant variety (Table 1) were evaluated for adaptation, disease resistance and better fruit quality. The WorldVeg tomato entries were previously characterized for presence of genes conditioning TYLCD resistance (Ty genes) and bacterial wilt resistance genes Bwr-12 and Bwr-6 and several other disease resistances [16]. Some commercial hybrids and popular inbred entries were included in this study because of their good fruit appearance and potential good yield. ICRXINA, one of the inbred varieties in this study, is already popular in Mali and other neighboring countries for its yield and heat resistance. Seedlings of these entries and varieties were raised in the nursery at the Samanko Research Station, WorldVeg during the rainy season (June to October 2019) and dry cool season (November 2019 to March 2020).

The field was plowed by a tractor, levelled and plot layout was prepared. Locally prepared compost from cow dung and crop residues at the rate of 15 t/ha was incorporated into the plots five days prior to transplanting. Seedlings in rainy season and dry season trials were transplanted at the 4–5 leaf stages. Each entry/variety was planted in two rows of six-meter length with a spacing of 50 cm between and within rows. Each row included 12 plants with 24 plants per plot. The treatments were laid out in a randomized complete block design with two replications. A normal agronomic practice, including periodic weeding and NPK (17-17-17) fertilization at the rate of 200 kg/ha after seedling establishment, was applied. Plants were watered as needed by drip irrigation. No synthetic or organic pesticides were applied to control diseases, insects and weeds.

**Table 1.** Tomato entries evaluated for disease reactions, fruit yield and quality in rainy and dry seasons, Bamako Mali.

| Distribution Code | Internal Code | Type | TYLCD Resistance Genes | | | Bacterial Wilt Genes | | | | | | RKN | FW | LB |
| | | | Ty1/3 | Ty-2 | Ty-5 | Bwr-12 | Bwr-6a | | Bwr-c | | Bwr6-d | Mi | I2 | Ph-3 |
| | | | | | | | 6–124 | 6–118 | 6–17 | 6–94 | 6–110 | | | |
| AVTO1003 | CLN3125L | Introduced Line | + | + | − | + | − | − | − | − | − | − | − | − |
| AVTO1007 | CLN3078A | Introduced Line | + | + | − | + | − | − | − | − | − | − | − | − |
| AVTO1008 | CLN3078C | Introduced Line | + | + | − | + | − | − | − | − | − | − | − | − |
| AVTO1122 | CLN3150A-5 | Introduced Line | − | − | + | + | − | − | − | − | − | − | − | − |
| AVTO1429 | FMTT1733D | Introduced Line | + | + | − | + | − | − | − | − | − | − | + | − |
| AVTO1464 | FMTT1733E | Introduced Line | + | + | − | + | − | − | − | − | − | − | + | − |
| AVTO1704 | CLN3900D | Introduced Line | + | + | − | + | − | − | − | − | − | + | + | − |
| AVTO1705 | CLN3902C | Introduced Line | + | + | − | + | − | − | − | − | − | − | + | + |
| AVTO1706 | CLN3961D | Introduced Line | + | + | − | + | − | − | − | − | − | − | − | − |
| AVTO1707 | CLN3961C | Introduced Line | + | + | − | + | − | − | − | − | − | − | − | − |
| AVTO1710 | CLN3641F | Introduced Line | − | − | − | + | + | + | + | + | + | − | − | − |
| AVTO1715 | CLN3938E | Introduced Line | + | − | − | + | + | + | + | − | + | − | − | − |
| AVTO1716 | CLN4018A | Introduced Line | − | − | − | + | + | + | + | + | + | − | − | − |
| AVTO1717 | CLN4018B | Introduced Line | − | − | − | + | + | + | + | + | + | − | − | − |
| AVTO1718 | CLN4018C | Introduced Line | − | − | − | + | + | + | + | + | + | − | − | − |
| AVTO1719 | CLN4018D | Introduced Line | + | − | − | + | − | − | + | + | + | + | − | − |
| AVTO1726 | CLN3902D | Introduced Line | + | + | − | + | − | − | − | − | − | − | + | + |
| AVTO1729 | CLN3961E | Introduced Line | + | + | − | + | − | − | − | − | − | − | − | − |
| H9205 | | Commercial hybrid (Heinz Seeds) | | | | | | | | | | | | |

**Table 1.** *Cont.*

| Distribution Code | Internal Code | Type | TYLCD Resistance Genes | | | Bacterial Wilt Genes | | | | RKN | FW | LB |
|---|---|---|---|---|---|---|---|---|---|---|---|---|
| | | | Ty1/3 | Ty-2 | Ty-5 | Bwr-12 | Bwr-6a | Bwr-c | Bwr6-d | Mi | I2 | Ph-3 |
| H9881 | | Commercial hybrid (Heinz Seeds) | | | | | | | | + | + | |
| ICRIXINA | | Popular inbred line variety | - | - | - | | | | | + | | |
| Kènèya | | Popular inbred line variety | | | | | | | | | | |
| Konica | | Popular inbred line variety | | | | | | | | | | |
| Nayeli | | Popular inbred line variety | | | | | | | | | | |
| UC82 | | Popular inbred line variety | | | | | | | | | | |
| VI043614 | H7996 | Bacterial wilt resistant rootstock | – | – | – | + | + | + + | + | | | – |

'+' homozygous for resistance allele, '-' homozygous susceptible allele, '/' = heterozygous, H = heterogeneous. Bwr-12 and Bwr-6 genes condition bacterial wilt resistance. Ty1/Ty3, Ty2 and ty5 genes condition resistance to tomato yellow leaf curl disease. Ph-3 conditions late blight (LB) resistance. Mi gene conditions resistance to the RKN = root-knot nematode (Meloidogyne incognita). I2 conditions resistance to race 2 of the FW = Fusarium wilt pathogen. WorldVeg lines prefix with AVTO code were evaluated for disease resistance genes. ICRIXINA is a pure line resistant to nematode and *Xanthomonas campestris* but susceptible to TYLCD. H9881 is a hybrid resistant to Verticillium wilt race 1, Fusarium wilt race 1 and 2 and nematode. Absence of + or – in the table means, no information is available if those commercial hybrids or popular inbred lines have resistance genes for one or more pathogens.

*2.3. Field Data Collection*

Data were collected from each row excluding one plant from both row ends. Plants were visually monitored at weekly intervals for the occurrence and symptoms of major diseases. Incidence values of target diseases were collected. The disease incidence (DI) was calculated as the proportion of infected plants per plot. The severity of bacterial leaf spot was scored using a 0–5 scale, where 0 is no disease, 1 = 20% of the leaf shows symptoms, 2 = 40% of the leaf shows symptoms, 3 = 60% of the leaf shows symptoms, 4 = 80% of the leaf shows symptoms and 5 is severe disease (death). Identification of diseases was based on the expression of symptoms at the field level. Observations of major insect pests were conducted but no data were collected.

Horticultural traits that were measured included plant height at flowering, number of days to 50% flowering (number of days from sowing until 50% of plants in the plot produced one flower), days to 50% fruiting, and days to first harvest (number of days from sowing to the first picking day). Fruit yield (t/ha) (marketable and non-marketable) of four harvests were collected excluding the two end plants of each row. In addition, daily and monthly air temperature, relative humidity and precipitation data were collected during the experimental period.

*2.4. Fruit Quality Analysis*

Ten relatively uniform sized and fully matured red-ripe fruits were collected from each of the tomato entries and varieties in the field and transported to the postharvest laboratory of the WorldVeg at Samanko Research Station for fruit quality (firmness, total soluble solid (TSS), pH and dry matter) measurement. Fruit firmness of those ten fruits of each entry/variety was determined using fruit penetrometer (FT011, Wagner Instruments, Milan, Italy). For each fruit, the penetrometer was applied to the equatorial axis at two different points and values were presented in Newtons (N). For the TSS measurement, fruits were ground using a blender at a high speed for one minute and the juice was extracted into a beaker using a cheesecloth. The TSS content (°Brix) was measured using a hand-held refractometer (Atago, Tokyo, Japan) by placing a few drops of the tomato juice on the reading prism of the refractometer. The extracted tomato juice was used for pH measurement with a pH meter (FiveEasy F20, Mettler Toledo, Greifensee, Switzerland).

Moisture content of the fruit samples was determined by the oven drying method [17]. About 10 g of chopped tomatoes were dried in a forced air-drying oven (Model BOV-T25F,

Biobase, Shandong, China) at 105 °C until constant weight. Thereafter samples were cooled in a desiccator containing silica gel for 2 h and the dry weight (Wd) was determined. The moisture content (mc) was determined using the formula mc (%) = 100[(Wi-Wd)/Wi] where Wi is initial weight. The dry matter content was obtained by subtracting the moisture content (mc) from 100.

*2.5. Data Analysis*

Disease incidence, yield and the different fruit quality data were subjected to analysis of variance (ANOVA) using GenStat v20 for Randomized Complete Block Design and treatment mean comparison was done by Tukey's honestly significant difference (HSD) at a 95% confidence interval in GenStat [18].

## 3. Results

*3.1. Disease Incidence*

The presence and incidence of major diseases varied with the season. TYLCD due to begomoviruses occurred in both seasons but average incidence among entries in the dry season (66.5%) was more than twice that of the rainy season (23.9%) (Tables 2 and 3). Dry weather favors higher whitefly populations, which transmit begomoviruses. Significant differences between entries for TYLCD incidence were detected in both seasons and entry mean incidence values ranged from 0–75%, and 32–100% in the wet and dry seasons, respectively. ICRIXINA recorded high TYLCD incidence in both dry (86%) and rainy (75%) seasons. Most WorldVeg entries (AVTO prefix) were homozygous for one or two Ty genes that condition TYLCD resistance. However, in the dry season trial, mean TYLCD incidence of entries with two resistance Ty genes and one Ty gene were 70% and 56%, respectively, suggesting that Ty genes did not offer sufficient protection and the presence of two Ty genes was not better than one Ty gene. Entries AVTO1704, AVTO1715 and AVTO1464 demonstrated relatively high TYLCD resistance during both seasons.

**Table 2.** Tomato entries/varieties performance during the rainy season of 2019 (June–September) at Samanko Research Station.

| Entry | %TYLCD | % Bw | BLS | Plt ht. (cm) | Days 50% Flowering | TY (t/ha) | NMY (t/ha) |
|---|---|---|---|---|---|---|---|
| AVTO1003 | 30.23 a–e | 25.0 ab | 1.42 e–j | 53.1 abc | 66.0 c | 7.60 a | 0.43 ab |
| AVTO1007 | 11.9 a–d | 18.8 ab | 0.81 a–d | 56.2 abc | 66.0 c | 17.1 a–d | 0.79 a–d |
| AVTO1008 | 19.1 a–d | 21.9 ab | 0.63 abc | 51.3 abc | 66.0 c | 19.3 a–d | 0.14 a |
| AVTO1122 | 29.2 a–e | 18.8 ab | 1.79 h–l | 52.9 abc | 64.5 abc | 24.7 b–e | 1.85 b–e |
| AVTO1429 | 16.5 a–d | 6.25 a | 1.22 d–g | 53.1 abc | 66.0 c | 25.9 b–e | 0.72 a–d |
| AVTO1464 | 0 a | 21.9 ab | 1.12 c–f | 59.3 bc | 63.0 ab | 25.4 b–e | 0.91 a–d |
| AVTO1704 | 9.38 abc | 15.6 ab | 1.73 g–l | 48.0 ab | 64.5 abc | 6.50 a | 0.36 ab |
| AVTO1705 | 8.33 abc | 28.1 ab | 0.37 a | 59.9 bc | 63.0 ab | 9.20 ab | 0.11 a |
| AVTO1707 | 49.6 b–e | 21.9 ab | 1.47 e–k | 49.3 abc | 64.0 abc | 17.4 a–d | 0.61 abc |
| AVTO1710 | 20.0 a–d | 9.38 ab | 1.94 jkl | 57.7 bc | 62.5 ab | 40.9 e | 2.51 e |
| AVTO1715 | 0 a | 15.6 ab | 1.19 c–g | 57.2 bc | 64.5 abc | 19.3 a–d | 1.00 a–d |
| AVTO1716 | 20.6 a–d | 25.0 ab | 1.40 e–j | 55.5 abc | 62.5 ab | 10.9 ab | 1.42 a–e |
| AVTO1717 | 9.38 abc | 15.6 ab | 1.60 e–k | 58.0 bc | 62.5 ab | 25.7 b–e | 2.21 de |
| AVTO1718 | 3.85 ab | 25.0 ab | 1.33 d–h | 54.0 abc | 63.5 abc | 24.9 b–e | 1.99 cde |
| AVTO1719 | 6.25 abc | 0 a | 1.35 d–i | 60.1 c | 65.0 bc | 26.3 b–e | 0.77 a–d |
| AVTO1729 | 56.4 de | 21.9 ab | 1.67 f–k | 55.8 abc | 64.0 abc | 12.9 abc | 0.33 a |
| H9205 | 26.7 a–d | 6.25 a | 2.01 kl | 51.5 abc | 62.5 ab | 28.8 cde | 0.59 abc |
| H9881 | 50.0 cde | 21.9 ab | 1.20 d–g | 52.4 abc | 62.0 a | 31.0 de | 0.60 abc |
| ICRIXINA | 75.0 e | 6.25 a | 1.90 i–l | 51.6 abc | 62.0 a | 38.9 e | 1.95 cde |
| Konica | 29.4 a–e | 37.5 b | 2.274 l | 53.9 abc | 62.5 ab | 14.4 a–d | 0.68 abc |

**Table 2.** *Cont.*

| Entry | %TYLCD | % Bw | BLS | Plt ht. (cm) | Days 50% Flowering | TY (t/ha) | NMY (t/ha) |
|---|---|---|---|---|---|---|---|
| Nayeli | 35.6 [a–e] | 9.38 [ab] | 1.05 [b–e] | 44.4 [a] | 63.5 [abc] | 25.6 [b–e] | 0.47 [abc] |
| VI043614 | 18.8 [a–d] | 15.0 [ab] | 0.54 [ab] | 58.6 [bc] | 63.0 [ab] | 25.3 [b–e] | 1.13 [a–e] |
| F-test (P) | <0.001 | 0.004 | <0.001 | <0.001 | <0.001 | 0.003 | <0.001 |

Means within the same column followed by the same letter(s) are not significantly different at $p \leq 0.05$. VI043614 = Hawaii 7996. TYLCD = tomato yellow leaf curl diseases, %BW = percentage of wilted plants due to bacterial wilt, BLS = bacterial leaf spot scored using a 0 to 5 scale, TY = total yield (marketable + unmarketable) and NMY = non-marketable yield.

**Table 3.** Tomato entries/varieties performance during the dry season of November 2019–March 2020 at Samanko Research Station.

| Entry | %TYLCD | Days 50% Flowering | Days 50% Fruiting | TY (t/ha) | NMY (t/ha) |
|---|---|---|---|---|---|
| AVTO1003 | 79.7 [d–g] | 57.0 [b–f] | 65.3 [c–g] | 9.72 [a] | 1.38 [abc] |
| AVTO1007 | 77.3 [d–g] | 56.0 [a–f] | 64.0 [b–f] | 11.0 [ab] | 1.53 [abc] |
| AVTO1008 | 56.9 [a–f] | 55.0 [a–e] | 62.0 [a–e] | 16.6 [ab] | 3.29 [d] |
| AVTO1122 | 62.3 [a–g] | 54.0 [a–d] | 62.0 [a–e] | 11.9 [ab] | 1.07 [abc] |
| AVTO1429 | 65.4 [a–g] | 54.0 [a–d] | 61.0 [a–d] | 10.8 [ab] | 1.31 [abc] |
| AVTO1464 | 45.0 [a–d] | 58.0 [b–f] | 64.0 [b–f] | 10.1 [a] | 0.55 [a] |
| AVTO1704 | 31.6 [a] | 61.0 [def] | 70.3 [fg] | 10.0 [a] | 1.48 [abc] |
| AVTO1705 | 64.5 [a–g] | 58.0 [b–f] | 65.0 [bf] | 8.20 [a] | 0.99 [ab] |
| AVTO1706 | 100.0 [g] | 50.7 [ab] | 56.0 [a] | 10.1 [a] | 1.56 [abc] |
| AVTO1707 | 94.4 [fg] | 52.7 [abc] | 58.3 [abc] | 10.9 [ab] | 1.96 [a–d] |
| AVTO1710 | 63.8 [a–g] | 53.0 [abc] | 60.0 [a–d] | 12.7 [ab] | 2.79 [cd] |
| AVTO1715 | 37.5 [abc] | 59.0 [b–f] | 66.3 [d–g] | 13.7 [ab] | 2.45 [acd] |
| AVTO1716 | 35.3 [ab] | 55.0 [a–e] | 63.0 [a–f] | 13.6 [ab] | 1.72 [a–d] |
| AVTO1717 | 52.4 [a–e] | 54.0 [a–d] | 62.0 [a–e] | 15.1 [ab] | 2.06 [a–d] |
| AVTO1718 | 70.0 [b–g] | 54.0 [a–d] | 59.3 [a–d] | 12.3 [ab] | 1.89 [a–d] |
| AVTO1719 | 68.1 [a–g] | 51.8 [ab] | 61.0 [a–d] | 10.2 [a] | 2.16 [a–d] |
| AVTO1726 | 64.4 [a–g] | 54.0 [a–d] | 65.0 [b–f] | 10.0 [a] | 1.16 [abc] |
| AVTO1729 | 87.5 [efg] | 54.0 [a–d] | 61.0 [a–d] | 11.4 [ab] | 2.14 [a–d] |
| ICRIXINA | 86.1 [efg] | 52.7 [abc] | 63.0 [a–f] | 13.3 [ab] | 1.36 [abc] |
| Kènèya | 59.7 [a–f] | 65.0 [f] | 73.0 [g] | 8.80 [a] | 0.83 [ab] |
| Konica | 67.9 [a–g] | 63.0 [ef] | 69.3 [efg] | 10.4 [ab] | 1.13 [abc] |
| Nayeli | 79.2 [d–g] | 61.0 [c–f] | 65.0 [b–f] | 15.9 [ab] | 1.47 [abc] |
| UC82 | 73.6 [c–g] | 57.0 [b–f] | 64.0 [b–f] | 15.8 [ab] | 1.13 [abc] |
| VI043614 | 73.6 [c–g] | 47.7 [a] | 57.0 [ab] | 20.3 [b] | 1.76 [a–d] |
| F-test (P) | 0.003 | <0.001 | <0.001 | 0.49 | 0.09 |

Means within the same column followed by the same letter(s) are not significantly different at $p \leq 0.05$. TY = total yield (marketable + unmarketable) and NMY = non-marketable yield.

Wilting due to bacterial wilt was observed in the rainy season trial. The result revealed significant differences among entries. Mean BW incidence among entries was 17.6%, and entry means ranged from 0–38%. All WorldVeg entries were homozygous for the bacterial wilt resistance gene Bwr-12 and six entries were also homozygous for all or parts of Bwr-6. Mean BW incidence of the nine entries homozygous for Bwr-12 and the seven entries homozygous for Bwr-12 and Bwr-6 was 20% and 15%, respectively, in the rainy season trial. H7996, the BW resistant line used in some areas as grafting rootstock showed 15% wilted plants. No wilted plants were observed for AVTO1719. Bacterial leaf spot (LBS) commonly occurs in tomato grown under high precipitation and temperatures of 24–30 °C [19], and spots can occur on leaves and fruit. All entries developed BLS symptoms, with the highest severity recorded on Konica (2.3) and H9205 (2.0). VI043614 (=H7996) and AVTO1705 developed the lowest BLS with mean scores of 0.5 and 0.4, respectively. Although the incidence and damage levels were not collected, tomato fruit borer (*Helicoverpa armigera*)

in both growth seasons and whiteflies during the dry season were major insect pests encountered in this trial.

### 3.2. Yield

The results showed significant differences among entries in the rainy season trial but not in the dry season trial. Significant mean yield among entries was highest in the rainy season trial (21.7 t/ha) compared to the dry season trial (12.2 t/ha). Entry means in the rainy season trial ranged from 6.5–40.9 t/ha. AVTO1710 (40.9 t/ha) produced the highest marketable yield in the rainy season trial, followed by ICRIXINA (38.9 t/ha). The total and unmarketable yield revealed significant differences among entries/varieties. Marketable yield in the dry season ranged from 8.2 t/ha (AVTO1705) to 20.3 t/ha (AVTO43614).

The number of days for 50% of the plants to flower was significantly different among entries and varieties in both of the seasons. However, there was no difference in the number of days for 50% fruit set in the rainy season, while there were differences among entries during the dry season. Plant height in the rainy season ranged from 44.4 cm for Nayeli to 60 cm for AVTO1719 (Table 2). However, entries and varieties during the dry season were stunted with a height ranging from 27 cm (AVTO1706) to 45 cm (VIO43614).

### 3.3. Fruit Quality

The entries and varieties grown during the dry season had higher TSS than in the rainy season. There were no significant differences in TSS values among entries/varieties in the rainy season, but the TSS content of AVTO1008 and AVTO43614 were higher while that of AVTO1464, AVTO1715 and AVTO1717 were the lowest. The TSS values of the harvested tomatoes in the rainy season were between 3.0 to 4.0° Brix. However, the TSS values during the dry–cool growing season ranged from 4.4 (VIO43614) to 7° Brix (AVTO1705 and Keneya). The lowest TSS during the dry season was higher than the highest TSS value measured during the rainy season and mean TSS of all entries/varieties during the rainy season was 3.43° Brix, while the mean TSS during the dry–cool season was 5.5° Brix (Figures 1 and 2).

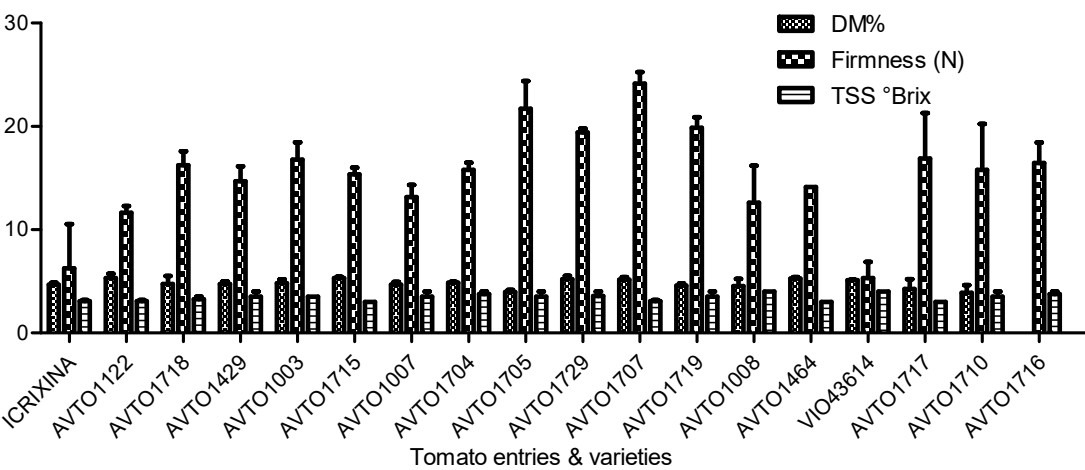

**Figure 1.** Postharvest fruit quality of tomato cultivated during the rainy season. DM = Dry matter and TSS = total soluble solids. Error bars indicated the standard error of means (S.E.M).

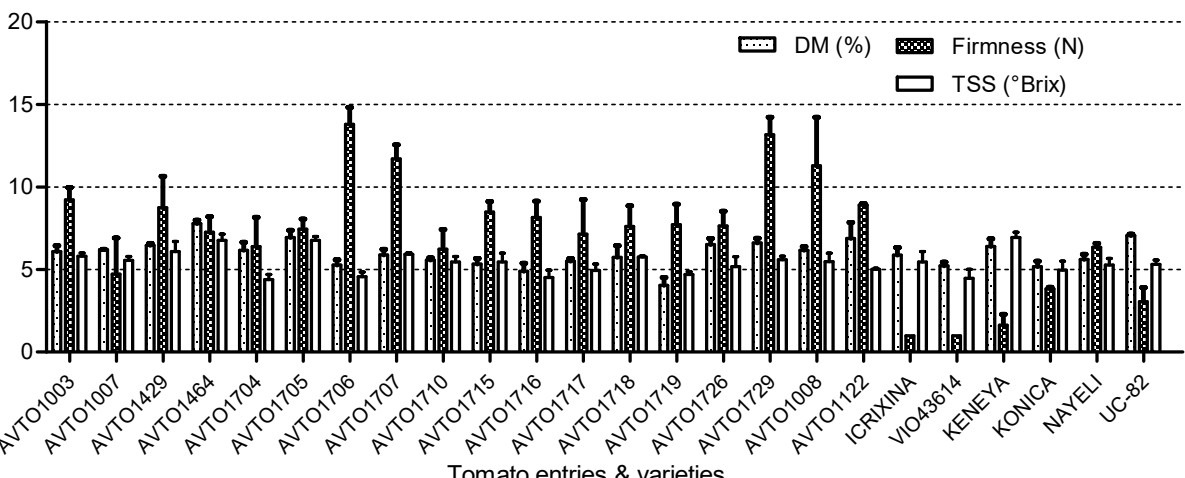

**Figure 2.** Postharvest fruit quality of tomato cultivated during the rainy season. DM = Dry matter and TSS = total soluble solids. Error bars indicated the standard error of means (S.E.M).

Significant differences ($p \leq 0.001$) among entries/varieties were found for fruit firmness and pH values (Figure 3). The mean pH during the rainy season was 4.68 (Figure 3a) and in the dry season was pH = 4.09 (Figure 3b). In the rainy season the pH values ranged from 4.3 in VIO43614 to 5.2 for AVTO1007, while in the dry–cool season the pH ranged from 3.89 in VIO3614 to 4.36 in AVTO1705. Fruit firmness and dry-matter content were different among entries/varieties, as well as seasons.

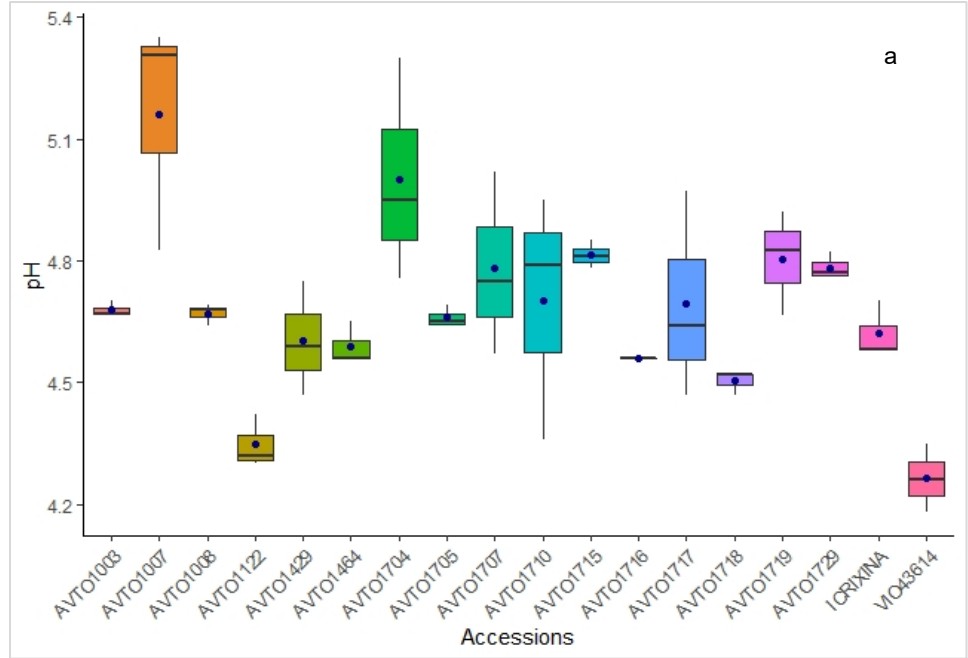

**Figure 3.** *Cont.*

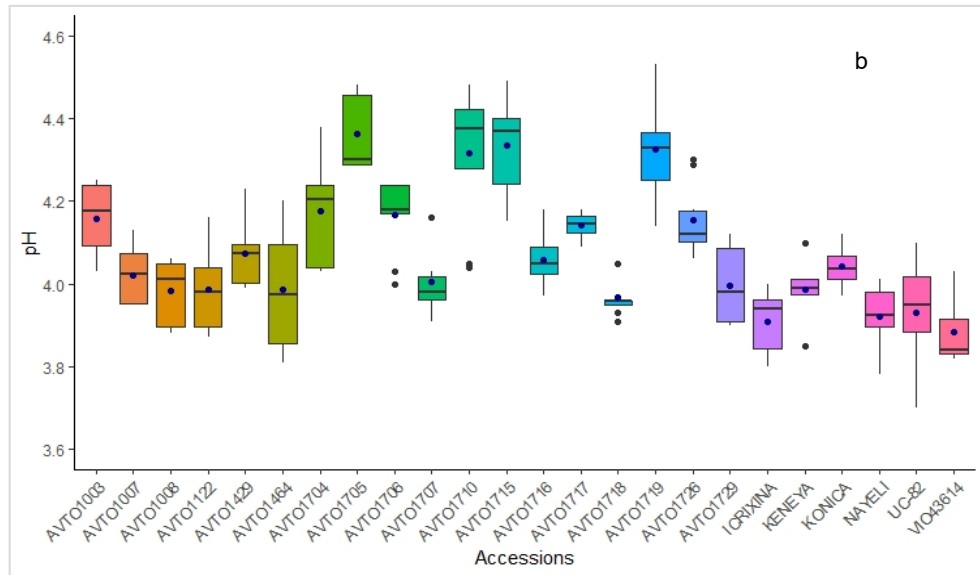

**Figure 3.** pH of tomato entries during rainy (**a**) and dry season (**b**) production consecutively with *n* = 10 fruits per entry. The line within each box indicates the median value of the data and the individual dots outside of the box are outliers.

## 4. Discussion

Tomato is one of the most important horticultural cash crops in Mali, where it can be produced in the dry and rainy seasons [4,8]. However, traditionally the peaks of production and availability in the market is during the dry period [4]. Due to different biotic and abiotic production constraints, farmers usually abandon tomato production during the unfavorable conditions. Hence, tomato in Mali is less available, its price is increased and consumption reduces during the hottest part of the dry season and the rainy season (April to September). This study identified tomato entries and varieties, which provided better yield and showing resistance to TYLCD and other diseases during the rainy season. Yield of these entries and varieties were low during the dry growing condition.

TYLCD incidence was high in both the dry and rainy season trials in this study, which is in agreement with previous studies [9,20,21], but severity was much higher in the dry season. The high viral pressure during the dry season in Mali (September to May) and other semi-arid agroecologies is mainly due to the high pressure of whitefly (*Bemisia tabaci*) vector populations [20] where the environment favors rapid reproduction and spread. Symptoms found in these trials included leaf curl/yellowing and curling, yellow mottle and crumpling and severe stunting and distorted growth. [11] In coordinated multilocation trials in Mali and six other West African countries, TYLCD severity was high and a major yield constraint in all countries; the molecular analysis of the samples taken from symptomatic plants identified three begomoviruses in Mali: tomato leaf curl Mali virus (ToLCMLV), tomato yellow leaf crumple virus (ToYLCrV) and tomato yellow leaf curl Mali virus. These three begomoviruses were also detected in TYLCD symptomatic plants in Burkina Faso, Ghana and Togo [21,22]. Similarly, a recent unpublished survey conducted in Mali identified three begomomoviruses (okra leaf curl virus (OLCV), pepper yellow vein Mali virus (PeYVMV) and pepper yellow vein virus (PeYVV)) infecting tomato using MiSeq $2 \times 300$bp V3 sequencing technology. In this study, the WorldVeg entries homozygous for two Ty genes as a group showed a slightly higher TYLCD incidence compared to those entries with a single Ty gene. It is possible that the Ty genes in these entries did not offer effective resistance to the begomoviruses encountered in the trials. Other studies identified several TYLCD resistant hybrids, including 'Atak' and 'Bybal', but none of these hybrids are currently available in Mali [11].

Bacterial wilt was not observed in the dry season trial but occurred extensively in the rainy season trial where it caused 38% wilting in the variety Konica. The bacterial wilt

pathogen is present in most of the vegetable production regions of Mali (Bamako, Sikasso, Koulikoro, Segoue and Kayes) with varying frequencies. A recent pathogen survey in Mali identified *Ralstonia pseudosolanacearum* isolates of phylotype I (Asian origin) and phylotype III (African origin) [10]. The same authors further characterized the bacterial wilt pathogen strains in Mali and identified four sequevars of Phylotype I and one sequevars of Phylotype III but sequevars 31 and 46 of phylotype I were the most frequent. Our results suggest that entries homozygous for bacterial wilt resistance genes Bwr-12 and Bwr-6 showed slightly higher BW resistance. Bacterial wilt disease caused by *Ralstonia pseudosolanacearum* was high during the rainy season due to the favorable temperature and soil moisture for the pathogen to develop and cause disease [10]. Generally, the study suggested tomato entries and varieties, which were adapted to local climate and resistant to diseases.

Significant differences were observed in yield and fruit qualities among entries and varieties in both the rainy and dry growing seasons. Tomato yields up to 40.9 t/ha were recorded during the rainy season in this study, which is normally a season characterized by production difficulties and low yield. However, during the dry season the total yields of the entries varied between 8 t/ha and 20 t/ha, which is comparable to the five years average national tomato yield [7]. Lower yield was recorded during the dry growing season, which could be due to the high incidence of viral diseases and the increased temperature of up to 41 °C. In addition to the pest and disease pressures, higher temperatures during the dry season led to reduced fruit set and lower yields contribute to a lower tomato yield on its direct effect on flower abortion and also indirectly make the tomato plant susceptible to diseases. The yields of ICRIXINA were relatively high in both dry and rainy season trials. ICRIXINA is among the popular variety in Mali and other countries in West Africa but susceptible to TYLCD. It was initially developed by ISRA (the Senegalese national agriculture research institute) for its better yield and was called XINA. However, its yield and quality were deteriorating through time. This XINA variety was later improved by the ICRISAT Niger office through five cycles of selection and purifications and the variety was renamed "ICRIXINA", which means XINA of ICRISAT [23].

Firmness, dry matter, TSS and pH are some of the quality attributes measured in this study. These quality parameters and others including fruit color, fruit size and shelf-life strongly influence consumers' purchasing and acceptance of the fruits. The mean dry matter content of 4.8% during the rainy season and 5.95% during the dry season lies within the 90–98% moisture content values of tomato fruits reported by [24]. Ref. [25] also reported that 95% of a ripe tomato is water. A variety of factors, including climatic and irrigation conditions influence tomato dry matter [25]. In this study, the high dry matter observed for fruit produced in the dry season could be due to the difference in the climatic and irrigation conditions between the dry and rainy seasons. Higher air temperatures promote the formation of the dry matter contents of tomato fruits [26,27]. Tomato fruit dry matter content is also influenced by the irrigation conditions with high irrigation levels associated with a decrease in the dry matter content [28]. In the rainy season, the exposure of the plants to rainfall may lead to excessive water supply and affects the dry matter content of the fruit unlike the dry season where plants were irrigated only through the drip system in a controlled manner.

The firmness of tomato entries and varieties coming from both the rainy and dry growing season varied significantly ($p = 0.05$). The observed variability in the firmness of the entries/varieties is related to the difference in their cell wall structure and composition, which is genetic-dependent [29]. On the other hand, the higher firmness values of the fruits could be attributed to their high dry matter [29–31] influenced by the difference in the climatic and irrigation conditions. Firmness is a good parameter to determine whether the tomato is for the processing or fresh market and if it can withstand a long-distance transport with better shelf-life. Processing tomatoes could change their color but maintain higher firmness ratings [32].

The differences between accession/varieties for TSS and pH are due to their individual genetic backgrounds. The TSS values of all entries except AVTO1008 and VIO43614 during

the rainy season were lower than previous reports from Ethiopia and Uganda [25,32]. Refs. [32,33] reported that TSS content is variety dependent and correlates negatively with tomato yield. Varieties with high TSS values tend to be less productive. This general statement agrees with the results of this study where tomato entries grown during the rainy season provided higher total yield but lower TSS content as compared to the tomatoes grown during the dry season, which recorded lower yields. The pH of tomato fruit is negatively correlated with the titratable acidity, which represents the organic acids (citric and malic acids) [34]. Entries/varieties with lower pH values (<4.5) are suitable for the tomato processing industry as lower pH prevents microbial proliferation and allows reduced energy and time during processing [35]. This study was conducted in a single wet growing and one dry growing season, but the result strongly suggested and supports other previous unpublished reports and observations in Mali.

## 5. Conclusions

Tomato farmers in Mali produce less productive, susceptible to pests and disease and lower quality tomato varieties. This study identified a better yielding of tomato entries or varieties adapted to the localities, with resistance to disease, better postharvest qualities and resilience to abiotic stresses. Tomato entry AVTO1710 provided high yield even in the rainy season where it is difficult to grow other local varieties. Dry season tomato production favors TYLCD due to the high pressure of the presence of insect vectors. There are entries that are moderately resistant to TYLCD during dry season production and highly resistant during the rainy season. Thus, these disease-resistant and high yielding tomato entries can be introduced to the IPM strategies and also in the breeding for disease resistant strategies in Mali. It is also recommended to choose a vector-free period of the year for reduced virus diseases pressure.

**Author Contributions:** Conceptualization, W.B. and R.S.; methodology, W.B., K.E.O. and J.-B.T.; investigation, W.B. and K.E.O.; data curation and analysis, W.B., K.E.O. and J.-B.T.; writing—original draft preparation, W.B. and K.E.O.; writing—review & editing, P.H., R.S. and K.N. All authors have read and agreed to the published version of the manuscript.

**Funding:** This work was supported by the long-term strategic donors to the World Vegetable Center (WorldVeg): Taiwan, the Foreign, Commonwealth & Development Office (FCDO) from the UK government, United States Agency for International Development (USAID), Australian Centre for International Agricultural Research (ACIAR), Germany, Thailand, Philippines, Korea, and Japan. It was also supported by the International Institute of Tropical Agriculture (IITA) through the West African USAID funded project named 'Africa Research in Sustainable Intensification for the Next Generation (Africa RISING)'.

**Informed Consent Statement:** Not applicable.

**Data Availability Statement:** The data presented in this study are available on request from the corresponding author. The data are not publicly available until the publication of this paper and later can be accessed from https://worldveg.tind.io/.

**Conflicts of Interest:** The authors declare no conflict of interest. We also declare that the funders had no role in the choice of research project; design of the study; in the collection, analyses or interpretation of data; in the writing of the manuscript; or in the decision to publish the results.

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
