# Peer review of "Evaluation of Different Tomato (Solanum lycopersicum L.) Entries and Varieties for Performance and Adaptation in Mali, West Africa"

_horticulturae, doi:10.3390/horticulturae8070579_

Round 1

Reviewer 1 Report

This manuscript is well written and an observational description of diseases in tomato varieties grown in Mali. This manuscript provides an interesting data set for farmers and would fit with MDPI Horticulturae 

Minor

'In contrary',  should be changed to 'In contrast' 

Author Response

Thank you very much for going through our manuscript.

The word was changed to "in contrast" as per your suggestion. 

Reviewer 2 Report

As a general comment, the manuscript could be really interesting because it illustrates the situation of tomato production in Mali in an extensive panorama (varieties used, climatic conditions and plant diseases and pests). Anyway, introduction have to be improved, mainly in the phytopathological topics. Moreover, the materials and methods paragraph must be integrated with more details about the recording of disease incidence and diagnostic analyses.

Due to the missing of line numbers, the comments are reported as follows (in bold, the comments):

Abstract

Twenty-two entries and varieties of tomato in the rainy season and 24 in the dry season were evaluated and well adapted, better yielder, diseases resistant and tomatoes with good fruit quality identified  - Re-phrase

Introduction:

Fusarium oxysporum f sp. oxysporum. This is not the name of the pathogen. Fusarium oxysporum f sp. lycopersici or Fusarium oxysporum f sp. radicis-lycopersici?

Verticilium dahlia: the correct name is Verticillium dahliae

Ralstonia sp.? : If the author doesn’t know which species of Ralstonia affected the fields, It could be better to write Ralstonia spp. (refer to EPPO description)

Materials and methods

In general, this paragraph must be improved. The materials and methods used to detect the resistance genes have to be described. The disease scale used for the symptoms observation must be reported for each pathogen and also the data analyses have to be reported.

The severity of bacterial leaf spot was scored using a 0-5 scale, where 0 is no disease and 5 is severe disease (death). The authors should describe all the level of the scale, from 0 to 5

Identification of diseases was based on the expression of symptoms at the field level and confirmed in the laboratory when necessary. Explain, somewhere in the text, the results of laboratory diagnoses (if available), especially for Fusarium and Verticillium wilt.

Horticultural traits that were measured included plant height at flowering, number of days to 50% flowerin. Change included with including

Results. The results cannot be evaluated due to missing information in materials and methods.

Discussion

Tomato leaf curl Mali virus (ToLCMLV), Tomato yellow leaf crumple virus (ToYLCrV) and Tomato yellow leaf curl Mali virus. These three begomoviruses were also detected TYLCD symptomatic plants in Burkina Faso, Ghana, and Togo. Add references

Author Response

We appreciate all your comments and suggestions which are invaluable to improve the quality of our paper. Following are our responses to each of the points.

Abstract

Twenty-two entries and varieties of tomato in the rainy season and 24 in the dry season were evaluated and well adapted, better yielder, diseases resistant and tomatoes with good fruit quality identified - Re-phrase!

  • Corrected! Sentence splitted into two to read as: “Twenty-two entries and varieties of tomato in the rainy season and 24 in the dry season were evaluated. Varieties which are well adapted, better yielder, diseases resistant and with good fruit quality identified”.

Introduction

Fusarium oxysporum f sp. oxysporum. This is not the name of the pathogen. Fusarium oxysporum f sp. lycopersici or Fusarium oxysporum f sp. radicis-lycopersici? Verticilium dahlia: the correct name is Verticillium dahliae

  • Corrected as: Fusarium oxysporum f.sp. lycopersici and Verticilium dahliae.

Ralstonia sp.? : If the author doesn’t know which species of Ralstonia affected the fields, It could be better to write Ralstonia spp. (refer to EPPO description)

  • Corrected as “Ralstonia species complex (RSC)”

Materials and methods

The materials and methods used to detect the resistance genes have to be described.

  • Unfortunately, the resistance genes indicated in Table 1 was from different studies of the World Vegetable Center and other sources.

 The severity of bacterial leaf spot was scored using a 0-5 scale, where 0 is no disease and 5 is severe disease (death). The authors should describe all the level of the scale, from 0 to 5

  • Edited according as: Severity of symptoms of bacterial leaf spot in tomato using a 0 to 5 rating scale as follows: 0 = healthy leaf, 1 = 20% of the leaf shows symptoms, 2 = 40% of the leaf shows symptoms, 3= 60% of the leaf shows symptoms, 4 = 80% of the leaf shows symptoms, and 5 = the complete leaf presents yellowing and starts to curl.

 Identification of diseases was based on the expression of symptoms at the field level and confirmed in the laboratory when necessary. Explain, somewhere in the text, the results of laboratory diagnoses (if available), especially for Fusarium and Verticillium wilt.

  • We think it is better to delete the statement “confirmed in the laboratory when necessary” since no major lab diagnosis were conducted as planned. Only sometimes, we did streaming test for Ralstonia spp. confirmation.  

Horticultural traits that were measured included plant height at flowering, number of days to 50% flowering. Change included with including

  • We think the word “included” looks ok.

 Discussion part

Tomato leaf curl Mali virus (ToLCMLV), Tomato yellow leaf crumple virus (ToYLCrV) and Tomato yellow leaf curl Mali virus. These three begomoviruses were also detected TYLCD symptomatic plants in Burkina Faso, Ghana, and Togo. Add references

  • Actually, the Leke et al 2015 was on the reference list but missed in the document. It is now referred as [21]. In addition to this reference the following reference is added in both the document and the reference list [22].
  • Sattar, M.N.; Koutou, M.; Hosseini, S.;, Leke, W,N.; Brown, J.K. Kvarnheden A First identification of begomoviruses infecting tomato with leaf curl disease in Burkina Faso. Plant Disease 2015, https://doi.org/10.1094/PDIS-08-14-0837-PDN.

 As a result of the addition of these 2 references, the reference number from 22 onward was shifted. The changes are indicated with track changes both in the document and reference lists.

Round 2

Reviewer 2 Report

Dear Authors, the revision made on the manuscript appears satisfactory, except for the answer given regarding the detection of the resistance genes that was carried out on the tomato entries. It is strongly recommended to report the methods used to obtain the results shown in table 1. It is suggested to add at least the references dealing with the different marker validation.

Author Response

Thank you very much for your important comments and suggestions.

We have added a reference Hanson et al. 2016 who screened and identified multiple disease resistant tomato entries associated with disease resistance genes.